# The anterior paired lateral neuron normalizes odour-evoked activity in the *Drosophila* mushroom body calyx

Luigi Prisco[1], Stephan Hubertus Deimel[2], Hanna Yeliseyeva[1], André Fiala[2], Gaia Tavosanis[1,3]*

[1]Dynamics of neuronal circuits, German Center for Neurodegenerative Diseases (DZNE), Bonn, Germany; [2]Department of Molecular Neurobiology of Behavior, University of Göttingen, Göttingen, Germany; [3]LIMES, Rheinische Friedrich Wilhelms Universität Bonn, Bonn, Germany

**Abstract** To identify and memorize discrete but similar environmental inputs, the brain needs to distinguish between subtle differences of activity patterns in defined neuronal populations. The Kenyon cells (KCs) of the *Drosophila* adult mushroom body (MB) respond sparsely to complex olfactory input, a property that is thought to support stimuli discrimination in the MB. To understand how this property emerges, we investigated the role of the inhibitory anterior paired lateral (APL) neuron in the input circuit of the MB, the calyx. Within the calyx, presynaptic boutons of projection neurons (PNs) form large synaptic microglomeruli (MGs) with dendrites of postsynaptic KCs. Combining electron microscopy (EM) data analysis and in vivo calcium imaging, we show that APL, via inhibitory and reciprocal synapses targeting both PN boutons and KC dendrites, normalizes odour-evoked representations in MGs of the calyx. APL response scales with the PN input strength and is regionalized around PN input distribution. Our data indicate that the formation of a sparse code by the KCs requires APL-driven normalization of their MG postsynaptic responses. This work provides experimental insights on how inhibition shapes sensory information representation in a higher brain centre, thereby supporting stimuli discrimination and allowing for efficient associative memory formation.

**\*For correspondence:**
Gaia.Tavosanis@dzne.de

**Competing interest:** The authors declare that no competing interests exist.

## Editor's evaluation

*Drosophila* Kenyon cells dendrites in the mushroom body calyx receive inputs from the projection neurons (PNs) in the antennal lobe. This work shows that potential variability of olfactory responses in Kenyon cell post synapses is reduced by the activity of a widely arborizing inhibitory interneuron named APL. APL also receives inputs from PNs and provides local scaled GABAergic feedback to PN-Kenyon cell synapses to normalize postsynaptic responses in the calyx.

## Introduction

Every day we are challenged to navigate through a complex and variable environment, often characterized by similar stimuli combined in different ways. Yet, our brain excels in assessing if, and how, the current experience is different or similar to a previously encountered one. The ability to discriminate across stimuli is achieved by minimizing the overlap between patterns of neuronal activity through a process defined as 'pattern separation' (*Santoro, 2013*). This conserved property is intrinsic to diverse circuits such as the mammalian cerebellum, the dentate gyrus, and the *Drosophila* mushroom body (MB) (*Cayco-Gajic and Silver, 2019*). In the current models, all the aforementioned circuits support pattern separation by utilizing different degree of inhibitory

mechanisms (*Tyrrell and Willshaw, 1992*; *Schweighofer et al., 2001*; *Sahay et al., 2011*; *Cayco-Gajic et al., 2017*; *Litwin-Kumar et al., 2017*). Experimental evidence in support of these inhibitory circuits has been described over the years (*Vos et al., 1999*; *Duguid et al., 2015*; *Inada et al., 2017*; *Parnas et al., 2013*; *Olsen et al., 2010*; *Lin et al., 2014*), however, the mechanism by which inhibition contributes to pattern separation is not yet fully understood, often due to technical limitations.

With an extended genetic toolkit and a brain of only ~100,000 neurons (*Raji and Potter, 2021*; *Alivisatos et al., 2012*) largely reconstructed at the EM level (*Zheng et al., 2018*; *Li et al., 2020a*), *Drosophila* represents an attractive system to provide experimental evidence on the mechanisms behind pattern separation. The fly MB receives mainly olfactory input, though optical, temperature, and humidity information is also represented (*Marin, 2020*; *Frank et al., 2015*; *Li et al., 2020b*). The MB is required for memory formation and retrieval (*Heisenberg et al., 1985*; *de Belle and Heisenberg, 1994*; *Dubnau et al., 2001*; *McGuire et al., 2001*; *Aso et al., 2014*). Within the MB input region, in the main calyx, olfactory projection neurons (PNs) deliver sensory information from 51 distinct olfactory glomeruli (*Grabe et al., 2016*; *Bates et al., 2020*) to ~2000 Kenyon cells (KCs) of the MB (*Aso et al., 2009*), for an expansion ratio of 40 (*Litwin-Kumar et al., 2017*). In the calyx, PNs synapse onto KCs via complex synaptic structures known as microglomeruli (MGs) (*Yasuyama et al., 2002*; *Kremer et al., 2010*; *Leiss et al., 2009*). At each MG, a single central PN bouton is enwrapped by, on average, 13 claw-like dendritic terminals of as many different KCs (Davi D Bock, personal communication). KCs integrate inputs in a combinatorial manner, with each KC receiving input from six to eight PNs, on average (*Butcher et al., 2012*; *Zheng et al., 2020*; *Li et al., 2020a*; *Turner et al., 2008*), of which more than half need to be coactive to elicit spikes (*Gruntman and Turner, 2013*). As a result, while PN odour-evoked activity is broadly tuned (*Perez-Orive et al., 2002*; *Bhandawat et al., 2007*), odour representation is sparse and decorrelated at the KCs layer (*Honegger et al., 2011*; *Turner et al., 2008*; *Campbell et al., 2013*; *Perez-Orive et al., 2002*), therefore reducing overlap between stimuli representation and allowing for better discriminability (*Kanerva, 1988*; *Cayco-Gajic et al., 2017*; *Olshausen and Field, 2004*). In addition to sparse PN:KC connectivity and KCs high input threshold, inhibition is required to reduce the overlap among odour representations in the *Drosophila* MB (*Lin et al., 2014*; *Lei et al., 2013*). At the MB, inhibition is provided by the GABAergic anterior paired lateral (APL) neuron, which innervates both the calyx and the lobes of the MB (*Liu and Davis, 2009*; *Pitman et al., 2011*; *Aso et al., 2014*). APL responds to odours with depolarization and calcium influx (*Liu and Davis, 2009*; *Papadopoulou et al., 2011*). Importantly, blocking APL output disrupts the KCs sparse odour representation and impairs learned discrimination of similar odours, pointing to its critical role in the process (*Lin et al., 2014*; *Lei et al., 2013*). APL is suggested to regulate sparse coding by participating in a closed feedback loop with the MB, similarly to its homolog giant GABAergic neuron (GGN) in the locust (*Papadopoulou et al., 2011*; *Lin et al., 2014*; *Litwin-Kumar et al., 2017*). However, APL is both pre- and postsynaptic to PNs and KCs in the adult calyx (*Yasuyama et al., 2002*; *Wu et al., 2013*; *Baltruschat et al., 2021*). Additionally, APL response to localized stimuli is spatially restricted (*Amin et al., 2020*). In particular, APL branches at the MB lobes and the ones in the calyx appear to represent two separate compartments (*Amin et al., 2020*), suggesting a possible distinct role of APL inhibition in these two different compartments. Hence, the mechanisms by which APL modulates sparse coding and its involvement in the process of pattern separation are still unclear. In the present work, we challenge the concept of a broad feedback inhibition to the MB calyx by APL with primary experimental data. In particular, we focused on the APL processes within the MB calyx and set out to identify the role of GABAergic inhibition at the PN:KC synaptic layer. Taking advantage of recently released EM datasets (*Scheffer et al., 2020*; *Zheng et al., 2018*), we report the complex synaptic interaction of APL with PNs and KCs within the MGs of the MB calyx. Next, via in vivo calcium imaging in the calyx, we explored the role of APL inhibition onto MGs by recording the odour-evoked activity of APL, PN boutons, and KC dendritic claws. Our results indicate that APL acts as a normalizer of postsynaptic responses to olfactory inputs in the MGs of the MB calyx, an idea that we confirmed by blocking the output of APL. Additionally, via volumetric calcium imaging, we addressed the locality of APL activation in the calyx and found that it is odour-specific. We suggest that the normalization of postsynaptic MG responses by APL is essential to determine the key property of KCs to respond only to the coincident input of PNs to multiple claws, allowing for an elevated stimulus discriminability.

## Results

### APL is an integral part of the microcircuit within MGs in the MB calyx

To better understand the role of GABAergic inhibition at the MB calyx, we investigated APL involvement into the calycal microcircuits with the highest resolution available. The APL of adult *Drosophila* innervates extensively all compartments of the MB, including calyx, lobes, and pedunculus (*Liu and Davis, 2009*). Moreover, the neuron appears to be non-polarized in the adult, with strong expression of both pre- and postsynaptic markers in all compartments (*Wu et al., 2013*). However, little is known regarding the detailed connectivity between APL and the cell types constituting the MB. Taking advantage of emerging electron microscopy (EM) datasets covering a full adult fly brain (FAFB, *Zheng et al., 2018*) or a large fraction of it (hemibrain, *Scheffer et al., 2020*), we examined the distribution of synaptic contacts between APL, PNs, and KCs, the major cell types constituting the MGs of the MB calyx (*Leiss et al., 2009*; *Yasuyama et al., 2002*; *Baltruschat et al., 2021*; *Figure 1A*). We recently reconstructed an entire MG in the FAFB dataset, starting from a PN bouton of the DA1 glomerulus and tracing all its pre- and postsynaptic partners (*Baltruschat et al., 2021*). Here, we focused on the synaptic connections involving APL. We found APL to be highly involved in the MG structure, with pre- and postsynaptic contacts with both KC dendrites and PN boutons (*Figure 1—figure supplement 1A*; *Baltruschat et al., 2021*). Many of those synapses were polyadic, displaying typical configurations within that specific MG (described in *Figure 1—figure supplement 1B*). To verify whether such features were specific to the DA1 MG reconstructed in *Baltruschat et al., 2021*, or common, we exploited the hemibrain EM dataset (*Scheffer et al., 2020*) and extracted all calycal connections from and to APL with either KCs or PNs. Out of the 136 PNs reported innervating the main calyx (*Li et al., 2020a*), 126 made and received synapses with APL (full list of PNs and APL interactions available at: https://doi.org/10.5061/dryad.bk3j9kdd1). To reveal the localization of these synapses, we rendered 3D graphs of single PNs derived from the hemibrain dataset (*Scheffer et al., 2020*) and mapped the synapses that they receive from APL within the MB calyx (see Materials and methods for details). Most of the APL-to-PN connections were localized on PN boutons (84% ± 2%, mean ± SEM, of the total synapses received by each PN localized on boutons), demonstrating that the majority of APL-PN interactions happens at MGs (*Figure 1B*, all images available at: https://doi.org/10.5061/dryad.bk3j9kdd1). Finally, to test whether the contribution of APL would be dependent on the input weight, defined as the total number of synapses received by APL from a particular input, we plotted the weight of APL-to-PN synapses against the PN-to-APL one. We found a positive correlation between the number of synapses made by APL towards a specific PN and the reciprocal synapses formed by that PN onto APL (*Figure 1D*). Of notice, most of the PNs not connecting to APL within the main calyx were already described as non-olfactory PNs (*Marin, 2020*; *Leiss et al., 2009*; *Li et al., 2020b*), and they all seemed to extend most of their terminals elsewhere, with little to no branches in the main calyx (*Figure 1—figure supplement 1D*). Similarly, of the 1919 KCs present in the dataset, 1871 displayed interactions with APL in the calyx. Mapping APL synapses onto single KC meshes (*Figure 1C*, all images available at: https://doi.org/10.5061/dryad.bk3j9kdd1) showed a majority of connections on KC claws. However, we noticed inhibitory synapses along KC dendrites as well. The KCs constituting the MB are divided into three major classes based on their axonal projections: γ, α/β, α'/β' (*Crittenden et al., 1998*; *Lee et al., 1999*). We found a difference in the spatial distribution of APL synapses depending on the KC type, suggesting that APL inhibition might have a different impact on different KC types. In particular, APL synapses onto α/β KCs were significantly less localized on claw-like dendritic terminals and more distributed along KC dendritic branches (*Figure 1—figure supplement 1C*, n = 210 [70 per KC type, randomly selected], p < 0.0001, unpaired ANOVA with multiple comparisons). The KCs not interacting with APL displayed a rather atypical structure, with extensive dendritic arborization just outside of the main calyx rather than within (*Figure 1—figure supplement 1E*). As in the case of PNs, the number of KC-to-APL synapses positively correlated with the APL-to-KC synapse number (*Figure 1E*). In conclusion, EM dataset analysis revealed a large involvement of APL in the calycal circuitry, with reciprocal connections to the vast majority of PNs and KCs. APL involvement in the MG structure as reported in *Baltruschat et al., 2021*, might be thus generalized to potentially all MGs of the main calyx.

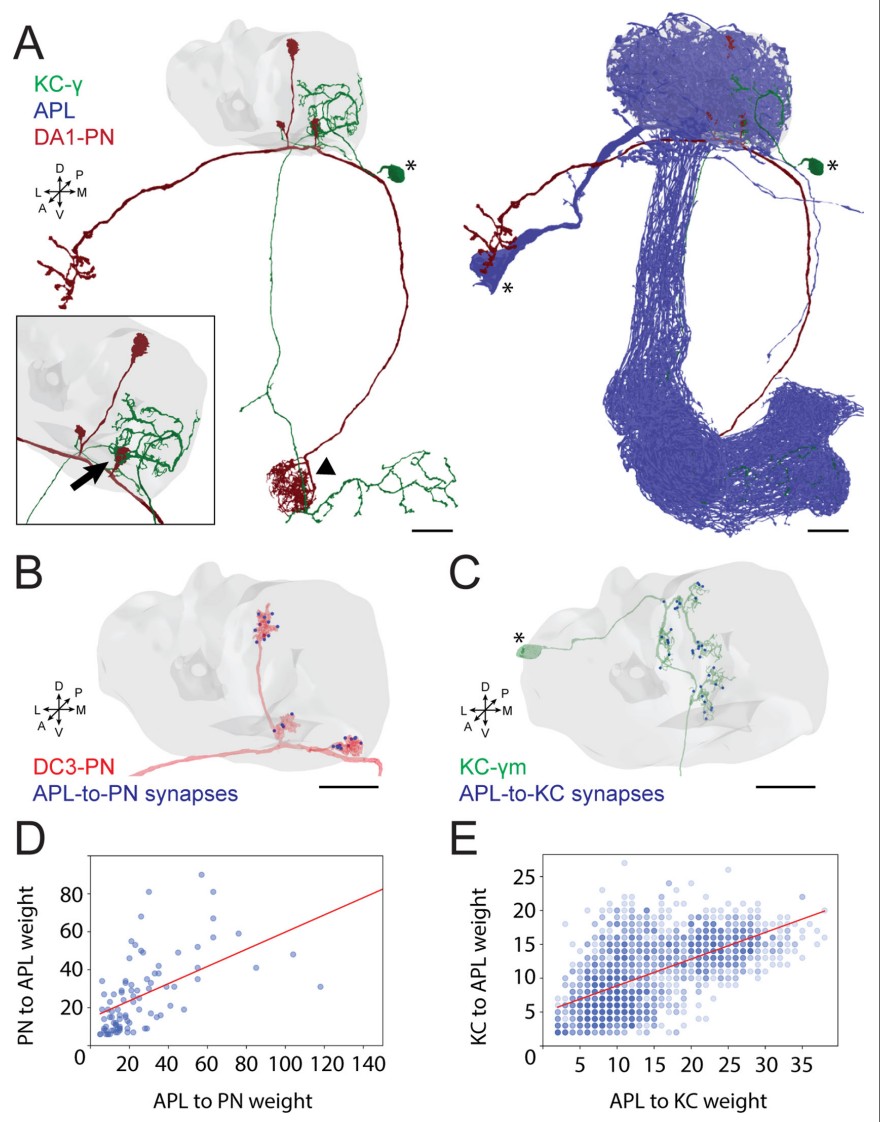

**Figure 1.** Anterior paired lateral (APL) participates in the microglomerulus (MG) microcircuit with reciprocal synapses. (**A**) Left: example of a projection neuron (PN) (red) sending collateral boutons into the mushroom body (MB) calyx (grey volume), where it connects onto Kenyon cells (KCs) claws via synaptic MGs. For simplicity, only one KC is visualized here (green). Note that the partial overlap between PN and KC indicated by the arrowhead is an artefact due to this particular view: the processes of these two neurons are located at different depths in this region. Bottom-left box: magnification of a PN bouton interacting with a KC claw (black arrow). Right: APL (blue) innervates the entire MB including lobes, peduncle, and the calyx. Asterisks indicate cell bodies. Scale bar = 10 µm. Axes indicate the orientation of the reconstruction; D (dorsal), V (ventral), L (lateral), M (medial), A (anterior), P (posterior). (**B**) Visualization of APL synapses (blue dots) onto a PN 3D mesh within the MB calyx. Most connections are localized on PN boutons. Scale bar = 10 µm. (**C**) Localization of APL synapses (blue) on a KC 3D mesh within the MB calyx. While most are localized on dendritic claws, some connections along dendritic branches could be seen as well (see also S1C). The cell body is marked by an asterisk. Scale bar = 10 µm. (**D**) Correlation between the number of PN-to-APL reciprocal synapses ($r^2 = 0.63$) and KC-to-APL ones (**E**) ($r^2 = 0.60$). The correlation was calculated among the entire synaptic weight (i.e. the total number of synapses reported in the dataset) that individual PNs or KCs had with APL. The gradient of blue in both scatter plots indicates how many neurons share the same connectivity values (lighter blue for fewer, darker blue for more). All 3D plots were created via the Neuprint-python package (see Materials and methods).

The online version of this article includes the following figure supplement(s) for figure 1:

**Figure supplement 1.** Anterior paired lateral (APL) in the microglomerular circuit.

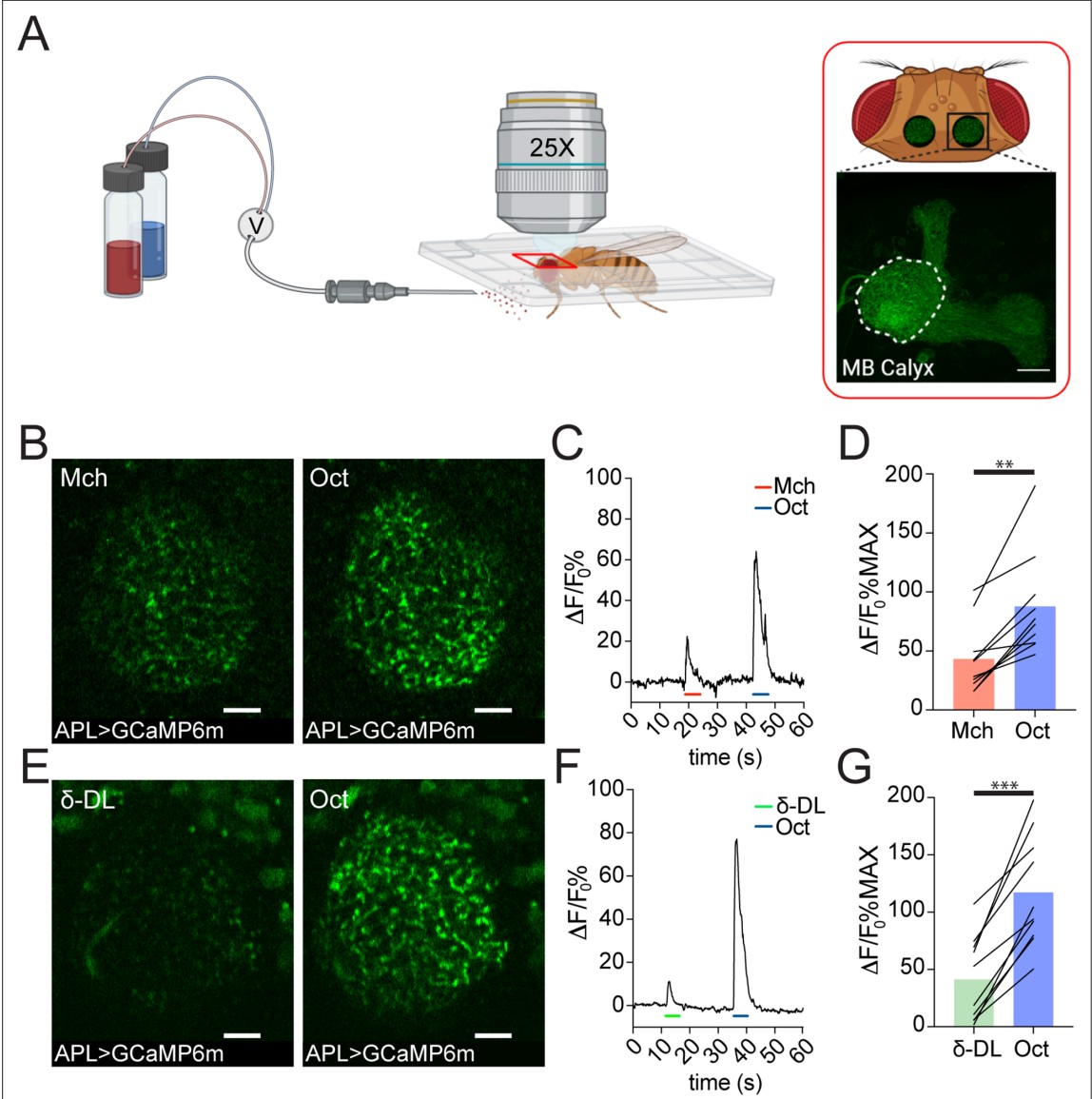

**Figure 2.** Anterior paired lateral (APL) responds to odours with variable calcium transients. (**A**) Schematic view of the two-photon in vivo imaging setup. Scale bar = 20 µm. (**B**) Example of APL response to 4-methylcyclohexanol (Mch) or 3-octanol (Oct) in the calyx of *APLi-GAL4> UAS-GCaMP6m* flies. Scale bar = 10 µm. (**C**) Fluorescence intensity over time for the fly showed in (**B**). (**D**) APL showed higher intracellular calcium transients in response to Oct compared to Mch. n = 10, p = 0.002 (**), Wilcoxon matched-pairs test. (**E**) Example of APL GCaMP6m response to δ-decalactone (δ-DL) or Oct. Scale bar = 10 µm. (**F**) Fluorescence intensity over time for the fly showed in (**E**). (**G**) APL peak response comparison for the δ-DL vs. Oct odours sequence. n = 10, p < 0.0001 (***), paired t-test. Odours were diluted 1:100, bars indicate means.

## In the calyx, APL displays different response levels to different odours

The analysis of the EM data provided structural evidence for possible feedforward and feedback circuits between APL, PNs, and KCs in the MB calyx (*Figure 1—figure supplement 1A-B*). To explore the functional role of APL in calycal MGs, we performed in vivo functional imaging experiments by expressing the calcium indicator GCaMP6m (*Chen et al., 2013*) specifically in APL via the APL intersectional *NP2631-GAL4, GH146-FLP* (APLi) driver (*Lin et al., 2014*; *Mayseless et al., 2018*) and recorded odour-evoked activity in the calyx (*Figure 2A*, see Materials and methods). Flies were stimulated with 5 s puffs of odours diluted 1:100 in mineral oil and exposed to sequences of two odours starting with 4-methylcyclohexanol (Mch) and 3-octanol (Oct), presented in a randomized fashion. Odour-elicited calcium transients in APL were detectable in the calyx (*Figure 2B and E*). Interestingly, we observed a clear difference in the GCaMP fluorescence levels, measured as ΔF/F$_0$ over the entire calycal region

innervated by APL (see also Materials and methods) in response to the two odour stimulations, with Oct eliciting a stronger APL response (*Figure 2C–D*, n = 10, p = 0.002, Wilcoxon matched-pairs test). To extend this observation, we exposed flies also to δ-decalactone (δ-DL), an odour reported to elicit the least overall activity in olfactory receptor neurons (ORNs) (*Hallem and Carlson, 2006*). Similarly, we measured a difference between the strength of the response to Oct compared to δ-DL (*Figure 2F–G*, n = 10, p < 0.0001, paired t-test). Moreover, the gap between the δ-DL signal peak and the Oct one was higher compared to the Mch vs. Oct group (Δ(Oct-Mch) = 45% ± 27%; Δ(Oct-δ-DL) = 76% ± 30%, n = 10, p = 0.0234, unpaired t-test with Welch correction), suggesting that APL is able to provide a variable, odour-tuned inhibition to MGs of the MB calyx.

## The response to different odours is highly variable in PNs, but more homogeneous in KC dendrites

To investigate the origin and the consequences of the observed difference in APL response at the MB calyx, we performed functional imaging experiments targeting the other two cell types participating in the microglomerular structure: PNs and KCs (*Leiss et al., 2009*; *Yasuyama et al., 2002*). Odours are detected by a large set of ORNs expressing chemically tuned odorant receptors (*Clyne et al., 1999*; *Hallem and Carlson, 2006*). ORNs project to the 51 distinct olfactory glomeruli in the adult antennal lobe (AL) in a stereotyped manner, with ORNs expressing the same odorant receptor projecting to the same glomerulus (*Gao et al., 2000*; *Vosshall et al., 2000*; *Grabe et al., 2016*). Within glomeruli, ORNs synapse onto second-order neurons, the PNs, which deliver odour information to higher brain regions such as the MB and the lateral horn (*Stocker et al., 1990*). To investigate whether odour-evoked activity in PNs reflected the differences in strength observed in APL, we expressed GCaMP6m in PNs via the generic PN-Gal4 driver *GH146* (*Berdnik et al., 2008*) and imaged PN dendrites in the AL. Flies were exposed to Mch or Oct as described for the APL imaging experiments, and the average peak among the responding glomeruli per brain was used as a general indicator of the total input transmitted by PNs. Imaging was performed on a single optical section of the AL, and only the glomeruli that could be unequivocally identified among all tested animals were taken into consideration for the analysis. While the number of responding glomeruli was similar between the two tested odours (*Figure 3—figure supplement 1*, also shown on a larger number of AL glomeruli in *Barth et al., 2014*), the overall calcium transient was higher when flies were exposed to Oct (*Figure 3—figure supplement 2A-B*, n = 10, p = 0.002, Wilcoxon matched-pairs test), suggesting that the main source of difference was represented by the degree of PNs activation rather than an additional/decreased number of active neurons. Likewise, a strong difference could be measured when flies were exposed to the δ-DL/Oct odours sequence (*Figure 3—figure supplement 2C-D*, n = 10, p < 0.0001, paired t-test), resembling the differences in APL activation detected at the MB calyx.

To address whether this odour-dependent variability in PN dendrites activity is still detectable within the collateral boutons in the MB calyx, we expressed the presynaptically localized GCaMP3 transgene *UAS-Syp::GCaMP3* (*Pech et al., 2015*) in PNs and recorded odour-evoked activity in PN boutons of the MB calyx. We exposed flies to Mch or Oct and calculated, per each odour trial, the average of the peak response among the boutons showing calcium transients (see Materials and methods for details). While the number of active boutons was similar between Mch and Oct stimulations (*Figure 3D*, n = 10, p = 0.689, paired t-test), the average response peak among active boutons was higher when flies were exposed to Oct (*Figure 3B*, n = 10, p = 0.0002, paired t-test). Furthermore, plotting the frequency distribution of all boutons activity peaks measured during these experiments showed a clear shift towards higher values of the entire Oct-responding population (*Figure 3C*, n = 10, p < 0.0001, Kolmogorov-Smirnov test). Hence, the difference shown in *Figure 3B* was not just due to a very high response of a few boutons, but rather to an overall increase in PN boutons activation levels across stimuli. Taken together, these data suggested that APL neuron activation scales with PN inputs strength. Indeed, while further odours will be needed to reach a more general conclusion, the overall level of PN activation elicited by Mch, Oct, or δ-DL correlated with the calycal APL response to those odours (Pearson's r correlation coefficient between APL activity in *Figure 2D and G* and PN activity in *Figure 3—figure supplement 2B*, D = 0.95).

To clarify the impact of the odour-tuned activation of APL on the response of KCs to odours, we next imaged the functional response of KC claws to odour stimulation. Flies expressing the postsynaptically tagged calcium indicator homer::GCaMP3 under the KCs promoter *MB247* (*Pech et al.,*

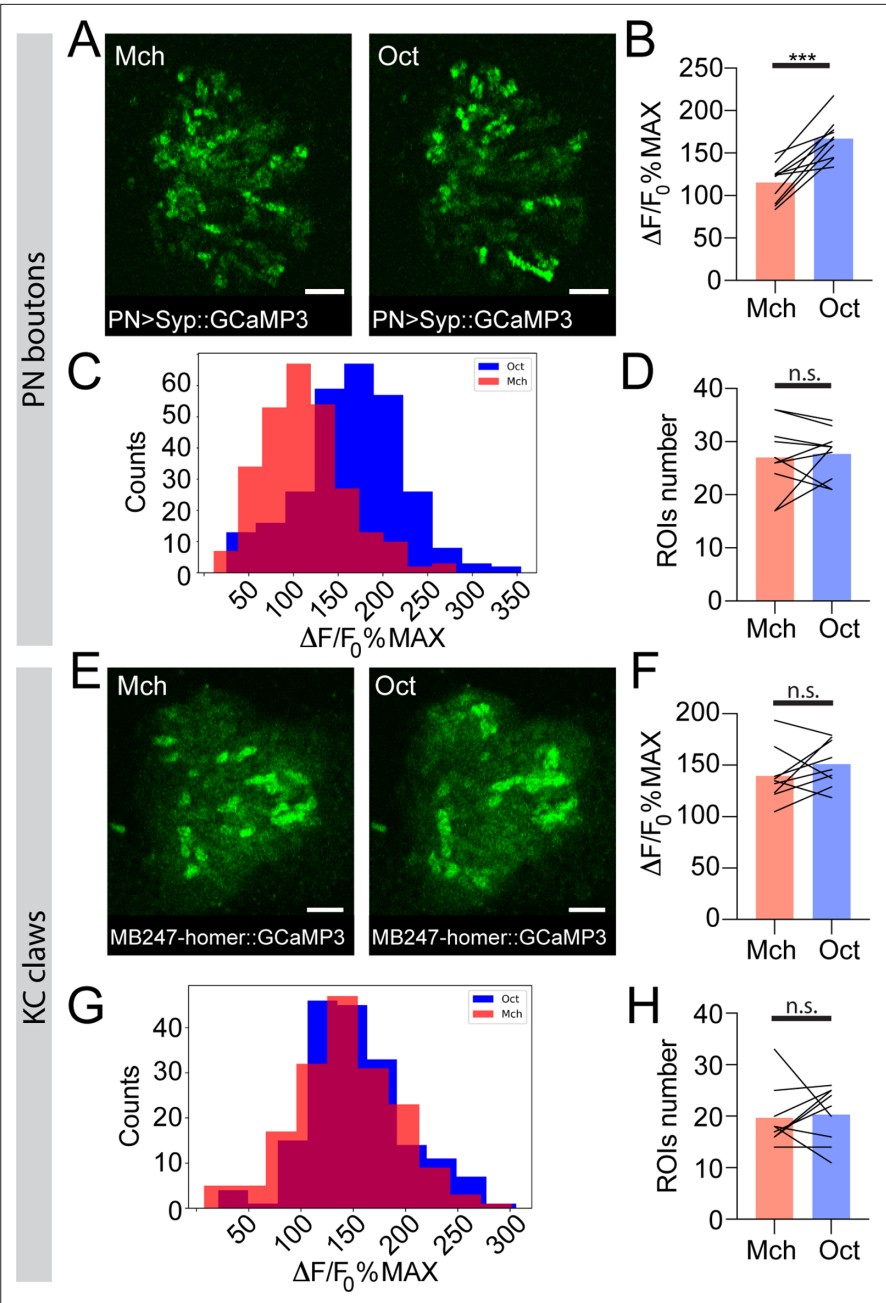

**Figure 3.** The strength of response to odour stimulation varies in an odour-dependent way in the projection neuron (PN) boutons, but is homogenous at the postsynaptic Kenyon cell (KC) claws. (**A**) Example of PN boutons fluorescence increase in response to stimulation with 4-methylcyclohexanol (Mch) or 3-octanol (Oct) in *NP225-GAL4> UAS-Syp::GCaMP3* flies. Scale bar = 10 μm. (**B**) The average activity peak among responding PN boutons was higher when flies were exposed to Oct compared to Mch. n = 10, p = 0.0002 (***), paired t-test. (**C**) Frequency distribution of PN boutons activity peaks in the Mch vs. Oct protocol. The Oct population was significantly shifted towards higher $\Delta F/F_0\%MAX$ values. n = 10, p < 0.0001 (***), Kolmogorov-Smirnov test. (**D**) The number of ROIs showing odour-evoked activity did not change between the two odour exposures. n = 10, p = 0.689, paired t-test. (**E**) Example of KC claws fluorescence levels in response to Mch and Oct in *MB247-homer::GCaMP3* flies. Scale bar = 10 μm. (**F**) The average activity peak among responding KC claws was comparable between Mch and Oct exposures. n = 9, p = 0.1648, paired t-test. (**G**) Frequency distribution of KC claws activity peaks in the Mch vs. Oct protocol. The two populations had a similar shape and spread among similar $\Delta F/F_0\%MAX$ values. n = 9, p = 0.0982, Kolmogorov-Smirnov test. (**H**) The number of ROIs showing odour-evoked activity did not change between the two odour exposures. n = 9, p = 0.727, Wilcoxon matched-pairs test. Odours were diluted 1:100, bars indicate means.

*Figure 3 continued on next page*

*Figure 3 continued*

The online version of this article includes the following figure supplement(s) for figure 3:

**Figure supplement 1.** Single projection neuron (PN) glomeruli responses at the antennal lobe (AL).

**Figure supplement 2.** Additional data on projection neurons (PNs) and Kenyon cells (KCs) odour-evoked activity.

---

*2015*) were prepared, stimulated, and imaged as described before. Olfactory stimulation caused the activation of different patterns of MGs in an odour-dependent manner (*Pech et al., 2015*; *Figure 3E*, *Figure 3—figure supplement 2E*). The number of MGs responding to each odour was not significantly different between Mch and Oct exposure (*Figure 3H*, n = 9, p = 0.727, Wilcoxon matched-pairs test), whereas it was lower when flies were stimulated with δ-DL (*Figure 3—figure supplement 2H*, n = 10, p = 0.0059, paired t-test), which elicits the least overall ORN activity (*Hallem and Carlson, 2006*) and induced a weak and restricted response in PN glomeruli at the AL (*Figure 3—figure supplement 1*). This difference in MG numbers might explain why in a previous report the total calycal functional response was higher for broader odours (*Apostolopoulou and Lin, 2020*). Importantly, the average postsynaptic response among MGs active in each odour trial was not different when comparing the response to Mch vs. Oct stimulation (*Figure 3F*, n = 9, p = 0.1648, paired t-test) or δ-DL vs. Oct stimulation (*Figure 3—figure supplement 2F*, n = 10, p = 0.767, Wilcoxon matched-pairs test). Additionally, the frequency distributions of the odour-evoked activity peaks were overlapping (*Figure 3G*, *Figure 3—figure supplement 2G*, n = 9–10, p = 0.0982 and p = 0.9554 for *Figure 3G* and *Figure 3—figure supplement 2G*, respectively, Kolmogorov-Smirnov test). Finally, the distance between the Oct- and Mch-induced activity in KC claws was significantly lower compared to the one measured in PN boutons (*Figure 4—figure supplement 1*, n = 10, p = 0.0271, Brown-Forsythe and Welch ANOVA test with multiple comparisons). Hence, the differences in activation strength described at the input population of MGs (the PN boutons) seemed to be normalized at the next neuronal layer, in the KC claw-like dendritic endings. Thus, the range of postsynaptic responses in MGs appears to be restrained.

## APL silencing leads to more variable odour evoked activity at the MGs postsynapses

Taken together, we showed that APL activation varies with different odours and scales with PN boutons activation levels. Together with the higher similarity among KC claws responses to different odours, this suggests a role of APL as normalizer of olfactory input-elicited response at the KC dendritic claws. If this is correct, blocking APL output would possibly lead to more variable activation of KC claws, mirroring PN bouton activation. We tested this hypothesis by expressing in APL tetanus toxin light chain (TNT), to block vesicle exocytosis and thus silence APL output (*Sweeney et al., 1995*). To suppress toxin expression during development, we co-expressed the temperature-sensitive GAL4 inhibitor *tubP-GAL80^ts^*. Flies were kept at 18°C until eclosion, and then transferred at 31°C for 24–48 hr prior to the experiments. KC responses in APL-silenced flies (APL OFF) were imaged with the postsynaptically tagged *MB247-homer::GCaMP3* construct. Due to the stochastic nature of *APLi-GAL4*-mediated expression (*Lin et al., 2014*), flies in which the flippase-dependent expression of TNT in APL did not happen were imaged as control (APL ON). Animals were stimulated with Mch or Oct. As expected, control animals did not show a difference in the average activity peak among responding MGs (*Figure 4B*, n = 10, p = 0.949, two-way ANOVA with Tukey's multiple comparisons) or in the number of responding units (*Figure 4D*, n = 10, p = 0.995, two-way ANOVA with Tukey's multiple comparisons). Furthermore, the frequency distributions of all MGs activity peaks measured in APL ON flies overlapped with each other (*Figure 4E*, n = 10, p = 0.0533, Kolmogorov-Smirnov test; see also *Figure 3G*). As previously reported, we observed a slight reduction in KC odour-evoked activity of control flies transferred at 31°C (e.g., compare *Figure 4B*, APL ON with *Figure 3F*), which was attributed to temperature sensitivity of either calcium binding, GCaMP3 conformation or of KC spiking dynamics (*Lin et al., 2014*). In calyces where the APL output was blocked by TNT expression though, we measured a significant difference in the response to the two tested odours, with Oct causing a stronger average odour-evoked activity (*Figure 4B*, n = 10, p = 0.0003, two-way ANOVA with Tukey's multiple comparisons) as well as a slight increase in the number of responding MGs (*Figure 4D*, n = 10, p = 0.047, two-way ANOVA with Tukey's multiple comparisons). Interestingly,

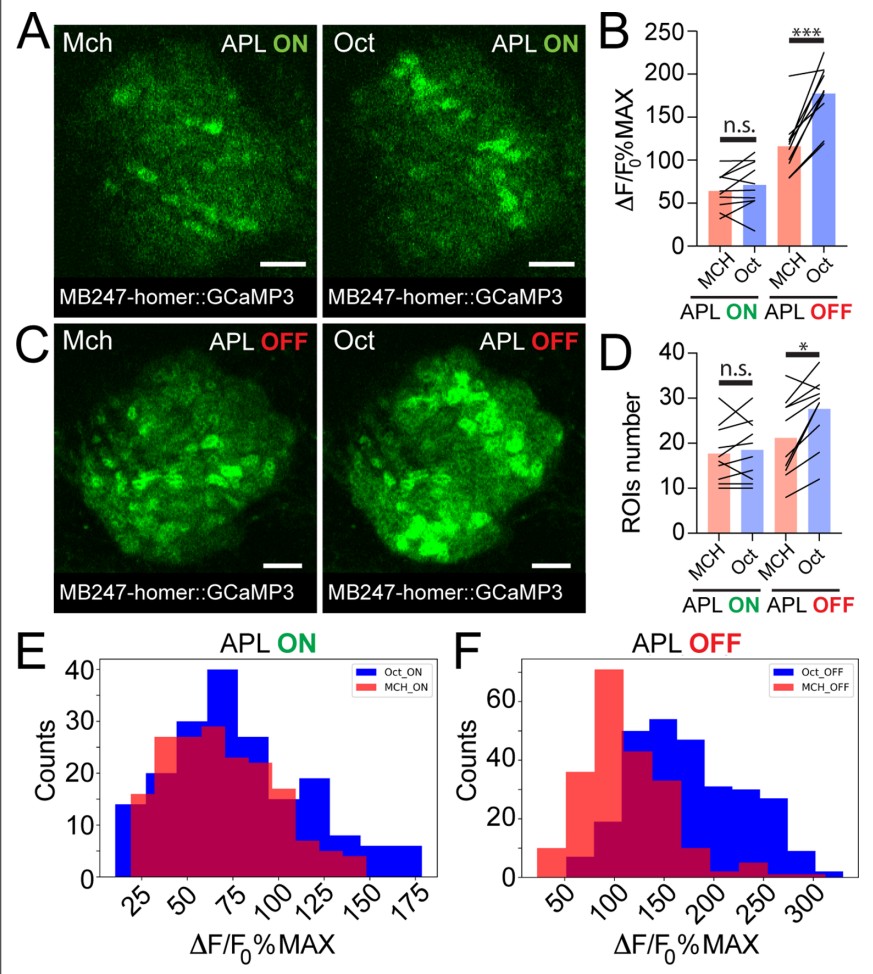

**Figure 4.** Anterior paired lateral (APL) silencing leads to more variable odour representations at the mushroom body (MB) calyx. Examples of Kenyon cell (KC) claws fluorescence levels in response to 4-methylcyclohexanol (Mch) and 3-octanol (Oct) in control animals (A, APL ON) or in flies where the output from APL was blocked (C, APL OFF). The genotype used is *APLi-GAL4> UAS TeTx, UAS-mCherry; MB247-homer::GCaMP3*. Scale bar = 10 µm. (**B**) The average activity peak in KC claws was similar when APL was active (APL ON, p = 0.949), but was highly variable in the absence of APL output (APL OFF, p = 0.0003 (***)). n = 10, two-way ANOVA with Tukey's multiple comparisons. (**D**) The number of odour-responding ROIs was comparable in the presence of active APL (APL ON, p = 0.995) and it was slightly increased in the absence of APL output (APL OFF, p = 0.047 (*)). n = 10, p = 0.047, two-way ANOVA with Tukey's multiple comparisons. (**E**) Frequency distribution of activity peaks among microglomeruli (MGs) responding to a particular odour in the presence of APL inhibition. The two populations are highly overlapping, as in *Figure 3G*. n = 10, p = 0.0533, Kolmogorov-Smirnov test. (**F**) In the absence of APL activity, the distribution of MGs responding to Oct shifted towards higher values, resembling presynaptic PN boutons data shown in *Figure 3C*. n = 10, p < 0.0001 (***), Kolmogorov-Smirnov test. Odours were diluted 1:100, bars indicate means.

The online version of this article includes the following figure supplement(s) for figure 4:

**Figure supplement 1.** Anterior paired lateral (APL) restrains odour-evoked activity in Kenyon cell (KC) claws.

small clusters of responding MGs were often found spatially close to each other in the APL OFF scenarios (e.g., compare spatial distribution of Oct responders in 4C vs. 4A) suggesting a possible locality of APL-mediated inhibition. Finally, the frequency distribution plot for the APL OFF flies showed two shifted populations, with Oct responses skewed towards higher values (*Figure 4F*, n = 10, p < 0.0001, Kolmogorov-Smirnov test), resembling the distribution displayed by the PN boutons (*Figure 3C*). Indeed, while the distance between the Oct- and Mch-induced activity was lower in KC claws of control flies in comparison to PN boutons (*Figure 4—figure supplement 1*, n = 10, p = 0.0024, Brown-Forsythe and Welch ANOVA test with multiple comparisons), no clear difference

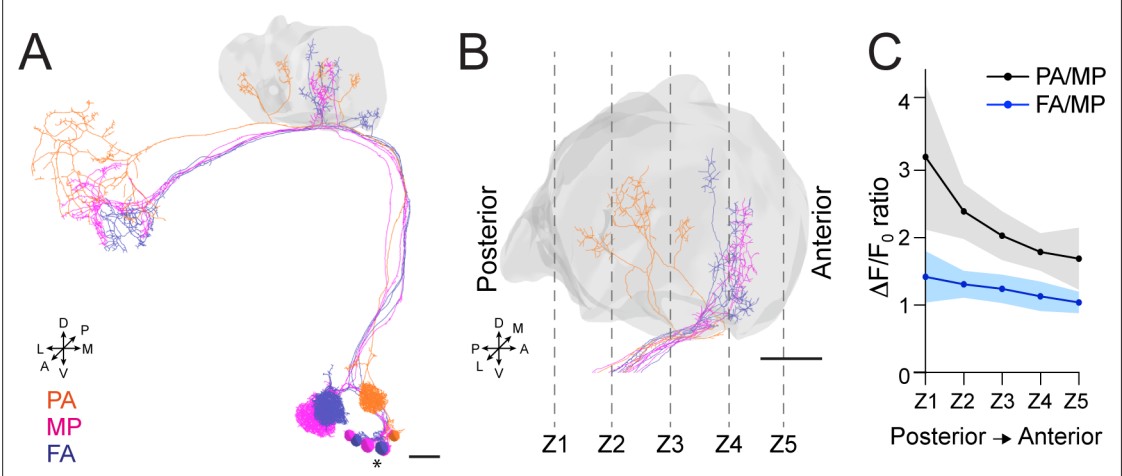

**Figure 5.** Anterior paired lateral (APL) inhibition is local within the mushroom body (MB) calyx. (**A**) 3D skeletons of the projection neurons (PNs) activated by the odours used in this experiment: pentyl acetate (PA) (orange), methyl palmitate (MP) (magenta), farnesol (FA) (blue). Asterisk indicates PN cell bodies. Scale bar = 10 μm. Axes indicate the orientation of the reconstruction; D (dorsal), V (ventral), L (lateral), M (medial), A (anterior), P (posterior). (**B**) Side view of the calyx showing the distribution of the PN terminals for the odours used in this experiment. Note the higher spatial segregation between PA and MP projection neuron (PN) terminals compared to FA and MP ones. The ticked lines Z1-Z5 show an example of sectioning applied when acquiring image stacks over time. Scale bar = 10 μm. (**C**) The APL neuron calcium transient ratio in response to PA vs. MP (black line) was highly variable across different sections of the calyx compared to the FA vs. MP one (blue line), and the slope of the two curves was significantly different. n = 7, p = 0.0004 (***), curves slope comparison via linear regression analysis. Coloured areas represent SEM.

The online version of this article includes the following figure supplement(s) for figure 5:

**Figure supplement 1.** Localized inhibition within the mushroom body (MB) calyx.

between PN boutons and KC claws could be measured in calyces where the output from APL was blocked (*Figure 4—figure supplement 1*, n = 10, p = 0.9698, Brown-Forsythe and Welch ANOVA test with multiple comparisons). Remarkably, KC responses recorded in the MB lobes were also higher and more diverse for different odours in the absence of APL inhibition (*Mittal et al., 2020*; *Apostolopoulou and Lin, 2020*; *Lin et al., 2014*).

In summary, blocking APL output led to a more variable odour representation at the level of KC claws. This variable odour representation bore a higher similarity to the activity of the input population, hence supporting the hypothesis that APL acts as an input strength normalizer on MGs in the MB calyx.

## APL inhibition onto MGs of the MB calyx is local

Our data indicate that APL contribution to the MG microcircuit yields a normalized postsynaptic response, independently of the variability of olfactory input strength. To understand how this input-tuned inhibition is achieved, we next asked whether the inhibition provided by APL is global and equally delivered to the entire calyx, or whether it might be more locally restricted to the sites of PN activation. Towards this aim, we selected three sets of PNs: DM3, VA1d, and DC3, which have distinctive bouton distributions in the calyx based on the hemibrain dataset (*Li et al., 2020a*). We expect the bouton distribution of these PNs in the calyx to be reproducible among animals (*Lin et al., 2007*; *Jefferis et al., 2007*). Based on an available database of odorant representations (*Münch and Galizia, 2016*), these PNs are activated by the following odours: Pentyl acetate (PA) activates the glomerulus DM3, whose PNs terminate in the posterior part of the calyx mainly; methyl palmitate (MP) activates the glomerulus VA1d, whose PN terminals populate the anterior part of the calyx; farnesol (FA) activates glomerulus DC3, with PN terminals that populate the anterior part of the calyx, similarly to VA1d (*Figure 5A–B*). We confirmed this regionalization of activities via volumetric imaging of calyces of *MB247-homer::GCaMP3* flies, which showed a prevalence of PA-responding MGs in the posterior part of the calyx, in comparison to MP and FA ones, respectively (*Figure 5—figure supplement 1A*, n = 7, p < 0.01 (**) for both PA vs. MP and PA vs. FA in Z1 and Z2, multiple t-tests). Taken together, this

combination of odours elicits reproducibly localized responses in the calyx, which can be segregated from one another, allowing to ask whether the response of APL might be local.

To test for a possible localization of APL activity throughout the calyx, flies expressing the calcium indicator GCaMP6m in the APL neuron were exposed to random sequences of the three odours, and 3D image stacks were acquired over time to define the localization of the APL response within the entire calycal volume (*Figure 5B*). Interestingly, we found the ratio of the calcium transients to be highly inconstant throughout the volume of the calyx for the structurally more distant odours PA and MP (see also Materials and methods for details). In particular, the PA/MP fluorescence ratio was higher in the posterior sections of the calyx and reduced in the anterior ones (*Figure 5C*, black line), reflecting the bouton distribution of the PNs activated by these two odours. By contrast, the fluorescence ratio of FA to MP was more homogeneous across the calyx volume, reflecting the fact that the boutons of the PNs that respond to these odours are located in a similar region of the calyx (*Figure 5C*, blue line). Of notice, we observed the same trend when comparing the ratio of PA to FA against the ratio of MP to FA (*Figure 5—figure supplement 1B*). In other words, not only APL calcium transients were different with different odours, as already shown (*Figure 2*), but these transients were also differently distributed in the calycal volume depending on the odour, strengthening the link between structural and functional data. Combined with the information about synapse distribution, these data suggest that APL is locally activated in the calyx at the MGs corresponding to activated PNs, and that the tone of APL inhibition in the calyx is a gradient that slowly degrades with increasing distance from the active boutons. Taken together, APL inhibition onto MGs of the MB calyx showed signs of locality. Of notice, a similar spreading mechanism has been observed in APL's parallel neurites at the MB lobes, where calcium transients failed to propagate over long distances (*Amin et al., 2020*).

## Discussion

While the importance of inhibition in reducing the overlap among stimuli representation has been postulated many decades ago (*Marr, 1969*; *Albus, 1971*; *Litwin-Kumar et al., 2017*; *Cayco-Gajic and Silver, 2019*) and supported by more recent experimental evidence (*Parnas et al., 2013*; *Olsen et al., 2010*), the complete mechanism by which inhibition supports stimuli discrimination is not fully understood yet. Here, we show that the inhibitory APL neuron, by participating in the structure of MGs of the *Drosophila* MB calyx, provides inhibition scaled to the PNs excitatory inputs to the calyx. As a result, the average strength and the distribution of postsynaptic responses in KC dendritic claws become more similar across different odour representations. We suggest that this normalization of postsynaptic responses operated by APL is at the core of pattern separation in the MB.

Pattern separation is obtained in the MB through the formation of a sparse response in the KC layer (*Honegger et al., 2011*; *Turner et al., 2008*; *Campbell et al., 2013*; *Perez-Orive et al., 2002*). The decoding of a sparse code, in general, increases the storage capacity of associative networks, thereby supporting learning and classification tasks (*Olshausen and Field, 2004*; *Kanerva, 1988*; *Tsodyks and Feigel'man, 1988*; *Vicente and Amit, 1989*; *Huerta et al., 2004*; *García-Sanchez and Huerta, 2003*; *Jortner et al., 2007*). In fact, sparse neuronal representations are described in several organisms including mammals, songbirds, and insects (*Rolls and Tovee, 1995*; *Vinje and Gallant, 2000*; *Hahnloser et al., 2002*; *Laurent, 2002*; *Quiroga et al., 2005*; *GoodSmith et al., 2017*; *Danielson et al., 2017*). APL was reported to play a key role in maintaining KCs responses sparse (*Lin et al., 2014*; *Lei et al., 2013*), but the underlying mechanism was far from understood. KCs receive inputs from six to eight PNs on average (*Butcher et al., 2012*; *Zheng et al., 2020*; *Li et al., 2020a*) and, due to KCs high firing threshold (*Turner et al., 2008*), require more than half of those inputs to be coactive to spike (*Gruntman and Turner, 2013*). Our data suggest that the APL neuron, by confining KC claws responses within a certain range of activation, ensures that KCs requirement of multiple coactive claws is respected even in the presence of highly variable input strengths. In other words, APL inhibition makes KC input integration dependent on the combinatorial pattern of inputs rather than on the strength of individual inputs. In support of this, blocking APL leads to an increased correspondence between input strength and KC response. Of notice, odour discrimination is achieved at multiple levels of the *Drosophila* olfactory pathway by different types of inhibitory neurons. Indeed, input gain control normalization has been described for GABAergic interneurons in the AL (*Olsen and Wilson, 2008*) as well as for inhibitory iPNs at the lateral horn (*Parnas et al., 2013*). Additionally, APL and its homolog GGN in the locust showed increased depolarization in response to increasing

odour concentration (*Papadopoulou et al., 2011*). This, combined with our findings, suggests that the normalization performed by APL might be acting not only across stimuli identities, but also among concentrations of the same stimulus.

Our structural and functional data point towards the involvement of APL in a feedforward loop from PN boutons to KC claws, as well as a closed feedback loop with PN boutons. An advantage of using recurrent circuits to provide inhibition is that such a system can deal with a wide range of input strength, as inhibition and excitation strengths are proportional. Indeed, EM analysis revealed both pre- and postsynaptic connection between APL and PN boutons, linearly proportional to each other (*Figure 1D*), and the differences in the APL calcium influx in response to odours correlated to the variability measured in PNs (*Figures 2 and 3*). So far, APL has been mainly described as a feedback neuron for KCs (*Lin et al., 2014*; *Lei et al., 2013*; *Amin et al., 2020*). However, feedforward inhibitory neurons from the input population onto the next layer have been described in other neuronal networks performing pattern separation (*Cayco-Gajic and Silver, 2019*). For example, granule cells receive both feedforward and feedback inhibition from Golgi cells at the cerebellar cortex (*Vos et al., 1999*; *Duguid et al., 2015*), which are driven by excitatory inputs from the mossy fibers (*Kanichay and Silver, 2008*) and granule cells' axons, respectively (*Cesana et al., 2013*). Moreover, it has recently been demonstrated that Golgi cells recruit scales with the mossy fibers input density (*Tabuchi et al., 2019*), similarly to what we observed in our functional imaging experiments. Additionally, adaptive regulation of KCs sparseness by feedforward inhibition has already been theorized in realistic computational models of insect's MBs (*Assisi et al., 2007*; *Luo et al., 2010*). Regarding KCs-to-APL connections, we found a positive linearity among pre- and postsynapses between these two cells (*Figure 1E*), confirming the presence of a local feedback loop within KC dendrites and the APL at the calyx (*Amin et al., 2020*). Furthermore, we reported that α/β KCs receive more inhibitory synapses along their dendritic trees compared to γ and α'/β', where the majority of synapses received from the APL is localized on KC claws instead (*Figure 1—figure supplement 1C*). As the ability of inhibitory synapses to shunt current from excitatory synapses depends on the spatial arrangement of the two inputs (*Spruston et al., 2016*), we speculate that the difference in APL synapses localization could contribute to some of the electrophysiological differences recorded among distinct KCs type. For example, $α/β_c$ KCs were found to have a higher input resistances and longer membrane time constants compared to α'/β' KCs, resulting in a sigmoidal current-spike frequency function rather than a linear one (*Groschner et al., 2018*). Additionally, a difference in synapses distribution can also indicate that two inhibitory mechanisms coexist at the MB calyx, similarly to what has been shown in the cerebellum where Golgi cells are responsible for both tonic inhibition, controlling granule cells spike number (*Brickley et al., 1996*), gain control (*Mitchell and Silver, 2003*), and phasic inhibition, limiting the duration of granule cells responses (*D'Angelo, 2009*).

Finally, volumetric calcium imaging showed that the APL inhibition is local within the MB calyx. In particular, we found a difference in the APL calcium transients when flies were stimulated with odours that activate PN subsets with segregated bouton distribution in the calyx (*Figure 5*, *Figure 5—figure supplement 1*). These data suggest that APL inhibition onto MGs can be imagined as a gradient that peaks at the MGs active during a given stimulus and attenuates with distance. Non-spiking interneurons in insects are typically large and characterized by complex neurite branching, an ideal structure to support local microcircuits (*Roberts and Brian M., 1981*). As a matter of fact, similar examples of localized APL response as described here have been reported in the *Drosophila* MB (*Amin et al., 2020*; *Wang et al., 2019*) as well as in the APL's homolog GGN in the locust (*Ray et al., 2020*; *Leitch and Laurent, 1996*). An advantage of having local microcircuits is that it allows a single neuron to mimic the activity of several inhibitory interneurons, as described in amacrine cells of both mammals (*Grimes et al., 2010*) and *Drosophila* (*Meier and Borst, 2019*). Additionally, a parallel local-global inhibition is suggested to expand the dynamic range of inputs able to activate KCs (*Ray et al., 2020*).

An important open question is whether the APL inhibition onto MGs of the MB calyx is more of a presynaptic phenomenon, therefore acting on PN boutons output, or a postsynaptic one on KCs claws. Our functional data reveal a clear impact of APL on the postsynaptic response in MGs (*Figure 3E–H*, *Figure 3—figure supplement 2E-H*, *Figure 4*), while the PN boutons display a broad range of activity levels (*Figure 3A–D*, *Figure 3—figure supplement 1*, *Figure 3—figure supplement 2A-D*). Accordingly, silencing of the $GABA_A$ receptor *Rdl* in KCs increased calcium responses in the MB, including the calyx (*Liu et al., 2007*), and reduced sparseness of odour representations (*Lei et al.,*

*2013*). However, due to the presence of presynapses from APL to PN boutons (*Figure 1B and D*), a presynaptic component of APL inhibition is certainly possible.

One possible caveat to our hypothesis is given by the fact that reducing GABA synthesis in APL by RNAi has been found to improve olfactory learning (*Lei et al., 2013*; *Liu and Davis, 2009*). However, this could be explained by a low efficiency of RNAi in this case. Indeed, incomplete silencing via RNAi increases KC output without affecting sparseness, whereas blocking APL output via *shibire*[ts] leads to large, overlapping odour representations and impaired olfactory discrimination (*Lin et al., 2014*).

Taken together, our study provides novel insights on how feedforward inhibition via APL shapes the postsynaptic response to olfactory inputs in the MB calyx and contributes to maintaining odour-evoked KC activity sparse. In the future, it will be interesting to investigate the impact of APL on memory consolidation, which has been associated with structural plasticity in the calyx (*Baltruschat et al., 2021*) and with changes in the KC response (*Delestro et al., 2020*).

# Materials and methods

**Key resources table**

| Reagent type (species) or resource | Designation | Source or reference | Identifiers | Additional information |
|---|---|---|---|---|
| Genetic reagent (*D. melanogaster*) | GH146-Gal4 | *Stocker et al., 1997* | BDSC:91812 | |
| Genetic reagent (*D. melanogaster*) | NP225-Gal4 | *Hayashi et al., 2002* | DGRC:112095 | |
| Genetic reagent (*D. melanogaster*) | NP2631-Gal4 | *Hayashi et al., 2002* | DGRC: 104266 | |
| Genetic reagent (*D. melanogaster*) | GH146-Flp | *Hong et al., 2009* | FLYB: FBtp0053491 | |
| Genetic reagent (*D. melanogaster*) | tubP-Gal80ts | *McGuire et al., 2003* | BDSC: 7017 | |
| Genetic reagent (*D. melanogaster*) | tubP-FRT-GAL80-FRT | *Gao et al., 2008*; *Gordon and Scott, 2009* | BDSC: 38880; 38881 | |
| Genetic reagent (*D. melanogaster*) | UAS-GCaMP6m | *Chen et al., 2013* | BDSC: 42750 | |
| Genetic reagent (*D. melanogaster*) | MB247-homer::GCaMP3 | *Pech et al., 2015* | | Gift from A Fiala |
| Genetic reagent (*D. melanogaster*) | UAS-Syp::GCaMP3 | *Pech et al., 2015* | FLYB: FBtp0130846 | |
| Genetic reagent (*D. melanogaster*) | UAS-TeTx | *Sweeney et al., 1995* | BDSC: 28838 | |
| Genetic reagent (*D. melanogaster*) | UAS-mCherry::CAAX | *Kakihara et al., 2008* | FLYB: FBtp0041366 | |

## Connectomics

Connectomics data were obtained from the hemibrain EM dataset (*Scheffer et al., 2020*) via the Neuprint analysis Tool (*Clements et al., 2020*). In particular, the Neuprint-python package (https://github.com/connectome-neuprint/neuprint-python) was used to filter for annotated synapses made and received by the APL only within the CA(R) ROI. The command *fetch_adjacencies* was used to extract data regarding the connectivity among cell types. To visualize neuron skeletons or filled renders in 3D, the commands *fetch_skeletons* or *fetch_mesh_neuron* were used instead. To visualize APL synapses onto PNs or KCs, the coordinates of those synapses were obtained via *fetch_synapse_connections* and plotted together with the 3D neuronal meshes. The localization of the synapses (e.g., on PN bouton or not) was addressed manually by two separate users in a blind manner, and the average counts were calculated. In particular, the localization of an APL-to-PN or APL-to-KC synapse was judged by looking at the filled 3D render of the neuron of interest. In case of uncertainties, both PN inputs and receiving KCs were plotted in the same image in order to directly identify the microglomerular structure. Detailed tables containing the list of all PNs and KCs interconnected with the APL within the MB calyx, as well as the weight of those synapses, and 3D images of APL synapses mapped onto PNs and KCs meshes can be found at: https://doi.org/10.5061/dryad.bk3j9kdd1.

## Fly strains

The following lines were used for experiments: *GH146-Gal4* (*Stocker et al., 1997*), *NP225-Gal4* (*Hayashi et al., 2002*), *NP2631-Gal4* (*Hayashi et al., 2002*), *GH146-Flp* (*Hong et al., 2009*), *tubP-GAL80*[ts] (*McGuire et al., 2003*), *tubP-FRT-GAL80-FRT* (*Gao et al., 2008*; *Gordon and Scott, 2009*), *UAS-GCaMP6m* (*Chen et al., 2013*), *MB247-homer::GCaMP3* (*Pech et al., 2015*), *UAS-Syp::GCaMP3*

(*Pech et al., 2015*), *UAS-TeTx* (*Sweeney et al., 1995*), *UAS-mCherry::CAAX* (*Kakihara et al., 2008*). Flies were raised in a 12 hr/12 hr light-dark cycle on a standard cornmeal-based diet at 25°C, 60% relative humidity unless they expressed the temperature-sensitive gene product Gal80ts. Flies carrying *tubP-GAL80ts* were raised at 18°C and placed at 31°C for 24–48 hr < 24 hr after eclosion. One- to seven-day-old flies were used for experiments. All experiments were performed on mixed populations of males and females.

## Two-photon in vivo calcium imaging

For in vivo imaging in the MB calyx, adult flies were briefly anaesthetized on ice, positioned in a polycarbonate imaging chamber (*Louis et al., 2017*), and immobilized using Myristic Acid (Sigma-Aldrich). To allow optical access to the Calyx, a small window was opened through the head capsule under Ringer's solution (5 mM HEPES, pH 7.4, 130 mM NaCl, 5 mM KCl, 2 mM CaCl$_2$, 2 mM MgCl$_2$; pH adjusted to 7.2). To minimize movement, fly heads were stabilized with 1,5% low melting agarose (Thermo Scientific) in Ringer's solution, immediately before dissection. Flies were imaged with a two-photon laser-scanning microscope (LaVision BioTec, TriM Scope II) equipped with an ultra-fast z-motor (PIFOC Objective Scanner Systems 100 μm range) and a Nikon 25× CFI APO LWD Objective, 1.1 NA water-immersion objective. GCaMP molecules were excited at 920 nm using a Ti:sapphire laser (Coherent Chameleon). Odours were delivered to the in vivo preparation via a 220A Olfactometer (Aurora Scientific) in a randomized fashion. Odours were loaded into the respective odour vials with a dilution 10× higher than the desired one, and further diluted 1:10 with clean air during odour stimulation. A constant flow of clean air was provided by a Stimulus Controller CS 55 (Ockenfels SYNTECH GbmH), equipped with two activated carbon inlet filters to avoid air contamination. Animals were stimulated with two odour puffs of 5 s each, separated by 20 s clean air intervals. Both clean air and odour flows were kept around 0.5 L/min for the entire experimental procedure. For imaging, a region large enough to include an entire z-section of the MB calyx was chosen. The scanning frequency was set around 9 Hz. Single plane videos were acquired unless stated otherwise.

For volumetric calcium imaging (*Figure 5*, *Figure 5—figure supplement 1*), flies were mounted and stimulated as described above and 3D stacks of five images each were acquired over time for the entire stimulation time. To compensate for the reduced speed caused by the stack acquisition, the frame rate was adjusted to around 16 Hz.

For in vivo AL imaging, female adult flies were briefly anaesthetized on ice, positioned in a custom-built fly chamber (*Hancock et al., 2019*), and immobilized using UV-hardening dental glue (Kentoflow, Kent Dental). A small dissection was performed in Ringer's solution (5 mM HEPES, pH 7.4, 130 mM NaCl, 5 mM KCl, 2 mM CaCl$_2$, 2 mM MgCl$_2$; pH adjusted to 7.2). Flies were imaged with a two-photon laser-scanning microscope LSM 7MP (Zeiss) equipped with a 20×/1.0 DIC M27 75 mm Plan-Apochromat objective (Zeiss). GCaMP molecules were excited at 920 nm using a Ti:sapphire laser (Coherent Chameleon). Odours were delivered to the in vivo preparation via a custom-built olfactometer. Odours were diluted in mineral oil (Sigma-Aldrich) at the required dilution. A constant flow of clean air was provided by a membrane pump 'optimal' (SCHEGO). Animals were stimulated with two odour puffs of 5 s each, separated by 20 s clean air intervals. Both clean air and odour flows were kept around 1 mL/s for the entire experimental procedure. For imaging, a region large enough to include an entire z-section of the MB calyx was chosen. The scanning frequency was set around 5 Hz. Single plane videos were acquired unless stated otherwise.

As the PN Gal4 driver *GH146* used in AL imaging experiments drives expression also in APL (*Liu and Davis, 2009*), the *NP225-Gal4* driver line (*Hayashi et al., 2002*) was chosen for PN boutons imaging at the calyx, as it targets a similar amount of PNs as *GH146* without including the APL.

## Data analysis

Two-photon images were analysed using Fiji (*Schindelin et al., 2012*). Briefly, raw videos were motion corrected via the 'Template Matching and Slice Alignment' ImageJ plugin (*Tseng et al., 2012*). Afterwards, ROIs of the single MGs responding to a given odour were automatically identified via the 'Cell Tracking by Calcium' ImageJ Macro (designed and written by DZNE IDAF). Briefly, given the expected ROI diameter (for our experiments, this value was set to 5 μm, in agreement with *Leiss et al., 2009*, and *Kremer et al., 2010*), this macro normalizes the whole dataset to the minimum image, obtained via ImageJ's 'Z-project' function, and thresholds the resulting data via a method of choice (default

used in this work: Otsu method, often used for analysis of dynamic fluorescence imaging data as in *Kaifosh et al., 2014*) in order to identify the active regions within a particular odour exposure. Hence, an ROI will be classified as active only if its pixel intensity variation is above threshold and its size matches the expected ROI diameter value. The output of the macro is a mask of active regions, which was then applied to the original, motion corrected video in order to extract the average intensity value over time per each of the detected ROIs. Finally, the $\Delta F/F_0\%$ and the $\Delta F/F_0\%MAX$ of each ROI was calculated by using the average intensity of the first 30 frames as $F_0$. Average activity peak values, plotted and compared in *Figures 3B, F and 4B* and *Figure 3—figure supplement 2F*, were obtained by averaging the peak response among all the active ROIs within each odour exposure trial. A detailed manual related to the ImageJ Macro, as well as Python notebooks computing the $\Delta F/F_0\%$ and $\Delta F/F_0\%MAX$ given a dataframe of intensity values over time, is available at: https://doi.org/10.5061/dryad.bk3j9kdd1.

To measure calcium influx among the APL neurites branching in the MB calyx, a manual ROI was drawn around the entire calycal region expressing the GCaMP and the average intensity value over time was extracted. $F/F_0\%$ and $\Delta F/F_0\%MAX$ were calculated as described above.

For the odours response ratio calculated in the APL volumetric calcium imaging experiment (*Figure 5*, *Figure 5—figure supplement 1*), the $\Delta F/F_0\%MAX$ per each odour was calculated as described above. The $\Delta F/F_0\%MAX$ values per each of the five frames contained in an image stack were obtained and averaged among all animals tested. Next, a $\Delta F/F_0\%MAX$ ratio between the odour pairs being compared was calculated and analysed per each of the sections included in an image stack.

## Confocal Imaging

To address the presence or not of the *APLi.GAL4*-driven *UAS-TNT* and *UAS-mCherry* products in the APL silencing experiment (*Figure 4*), whole flies were fixed on formaldehyde (FA) 4% in PBS with 0.1% Triton X-100 (PBT 0.1%) immediately after in vivo imaging. Once all animals sustained the in vivo imaging protocol, brains were dissected using a pair of forceps in a small Petri dish covered with a layer of silicon, fixed for further 20 min on FA 4% in PBT 0.1%, washed 3 times for 5 min ca in PBT 0.1% and mounted with Vectashield Plus Antifade Mounting Medium (Vectorlabs) on 76 × 26 mm microscope slides (Thermo Scientific) with 1# coverslips (Carl Roth). Brains were oriented with the dorsal part facing upwards. Imaging was performed on an LSM 780 confocal microscope (Zeiss) equipped with a Plan-Apochromat 63×/1.4 Oil DIC M27 objective (Zeiss). 512 × 512 pixels images were acquired, covering a region of the brain big enough to include the APL soma and branches around the MB calyx and MB lobes. Brains and their related in vivo imaging data were assigned to the classes 'APL OFF' or 'APL ON' based on the presence or not of the TNT-co-expressed mCherry fluorescence, respectively.

## Statistics

Statistics were carried out in Prism 8 (GraphPad). Parametric (t-test, ANOVA) or non-parametric tests (Wilcoxon, Mann-Whitney, Kruskal-Wallis, Kolmogorov-Smirnov) were used depending on whether data passed the D'Agostino-Pearson normality test. Statistical power analysis was conducted in G*Power (*Faul et al., 2007*). N values indicate number of flies hence biological replicates, unless stated otherwise. Tables containing source data per each graph can be found at: https://doi.org/10.5061/dryad.bk3j9kdd1.

## Acknowledgements

We thank LMF and IDAF sections at DZNE, Seth Tomchick, Sanjeev Kaushalya, and Kevin Kuepper for support in technical development; Rita Kerpen, Phuong Tran, and Olga Sharma for technical assistance. We thank the Bloomington Stock Center, Andrew Lin, and Oren Schuldiner for fly lines. We thank Andrew Lin, Martin Nawrot, Peter Kloppenburg for discussions. We thank Davi D Bock and Zhihao Zheng for help with electron microscopy data. We thank Moshe Parnas, David Owald, Barbara Schaffran, and the members of the Tavosanis lab for critical reading of the manuscript. This work was supported by the DFG FOR 2705 to GT.

## Additional information

### Funding

| Funder | Grant reference number | Author |
|---|---|---|
| Deutsche Forschungsgemeinschaft | FOR 2705 | André Fiala<br>Gaia Tavosanis |

The funders had no role in study design, data collection and interpretation, or the decision to submit the work for publication.

### Author contributions

Luigi Prisco, Conceptualization, Data curation, Formal analysis, Investigation, Methodology, Project administration, Visualization, Writing - original draft; Stephan Hubertus Deimel, Data curation, Formal analysis, Investigation, Methodology, Visualization, Writing - review and editing; Hanna Yeliseyeva, Formal analysis, Methodology; André Fiala, Funding acquisition, Resources, Supervision, Validation, Writing - review and editing; Gaia Tavosanis, Conceptualization, Funding acquisition, Project administration, Resources, Supervision, Validation, Writing - review and editing

### Author ORCIDs

Luigi Prisco (ID) http://orcid.org/0000-0002-5896-9346
Stephan Hubertus Deimel (ID) http://orcid.org/0000-0002-4678-4926
André Fiala (ID) http://orcid.org/0000-0002-9745-5145
Gaia Tavosanis (ID) http://orcid.org/0000-0002-8679-5515

### Decision letter and Author response

Decision letter https://doi.org/10.7554/eLife.74172.sa1
Author response https://doi.org/10.7554/eLife.74172.sa2

## Additional files

### Supplementary files

• Transparent reporting form

### Data availability

All source data used to generate figures are available at: https://doi.org/10.5061/dryad.bk3j9kdd1.

The following dataset was generated:

| Author(s) | Year | Dataset title | Dataset URL | Database and Identifier |
|---|---|---|---|---|
| Tavosanis G | 2021 | Data from: The anterior paired lateral neuron normalizes odour-evoked activity in the Drosophila mushroom body calyx | https://doi.org/10.5061/dryad.bk3j9kdd1 | Dryad Digital Repository, 10.5061/dryad.bk3j9kdd1 |

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
