## [Editor Report]

*Drosophila* Kenyon cells dendrites in the mushroom body calyx receive inputs from the projection neurons (PNs) in the antennal lobe. This work shows that potential variability of olfactory responses in Kenyon cell post synapses is reduced by the activity of a widely arborizing inhibitory interneuron named APL. APL also receives inputs from PNs and provides local scaled GABAergic feedback to PN-Kenyon cell synapses to normalize postsynaptic responses in the calyx.

---

## [Decision Letter]

**Decision letter after peer review:**

Thank you for submitting your article "The anterior paired lateral neuron normalizes odour-evoked activity in the *Drosophila* mushroom body calyx" for consideration by *eLife*. Your article has been reviewed by 2 peer reviewers, and the evaluation has been overseen by Mani Ramaswami as Reviewing Editor and K VijayRaghavan as the Senior Editor. The reviewers have opted to remain anonymous.

Essential revisions:

Please revise the manuscript as requested in the "Recommendations for the authors". As you will see, almost all these can be addressed by including additional qualifications, information or methods of analyses without need for new experiments. Please endeavour to respond to constructively to all of them, unless you have good reasons not to follow one or other specific suggestion.

Recommendations for the authors.

1. The data seem to support the authors conclusions that stronger odor responses in PNs/KCs correlates with stronger APL response and thereby stronger GABAergic feedback. Do correlations seen from comparison of APL response magnitudes for MCH and Oct (Figure 2 and Figure 3) hold true for other odor pair comparisons? If such (APL vs PN/KC) comparisons cannot be made, then the fact that studies of additional odorants are needed for more definitive conclusions should at least be discussed.

2. The main conclusion of the manuscript rests on the findings that in one condition (PN boutons, or KC claws with APL blocked), OCT>MCH, but in another condition (KC claws in wildtype), OCT=MCH. Currently this is shown using p<0.05 vs. p>0.05, but it needs to be shown with either an interaction in a 2-way ANOVA, or directly comparing the difference (Oct-MCH) between the conditions (as they did in comparing the differences Oct-dDL vs. Oct-MCH in Figure 2G vs. 2D) (see Nieuwenhuis et al., Nat Neurosci 2011).

3. Early figures (Figure 1A-C) need some labeling to orient nonspecialist readers, particularly directional markers (anterior/poterior, medial/lateral), similar to figure 5. Scale bars should also be included on these anatomical figures throughout.

4. The spatial analysis was done for 5 anterior-posterior zones (Z1-Z5), selected as horizontal slices through the calyx. This seems to make sense for most of the PN terminal fields, but what about the other two axes (medial-lateral, dorsal-ventral)? For instance, some PNs (e.g., the PA-responsive one) traverse more than one A-P slice and have terminal fields in multiple dorsoventral layers.

5. In figure 5, an analysis directly comparing the spatial patterns of APL with PN/KC responses would bolster the evidence for local inhibition in shaping the responses. 5. Does silencing APL alter the regionally-specific responses in KCs? Providing an experimental test would bolster the evidence for a local role of APL modulating these spatial patterns.

6. The authors quantify PN bouton and KC claw responses by selecting only ROIs that responded to the odor. (line 303-4: "calculated per calyx the average peak response among the boutons showing calcium transients") However, sometimes they refer to "average bouton response" or "average microglomerular postsynaptic response" which could reasonably be interpreted to mean the average including non-responsive boutons/claws. This point should be clarified either by changing the phrasing or adding more explanations of the quantification. It's also not clear if the responses in Figure 3B, 3F, 4B are quantified for the same ROIs across both odors, or if for each odor, only the ROIs responsive to that particular odor were used. Could the authors be more explicit about their quantification methods? In the Methods section, they should explain how they identified bouton/claw ROIs (e.g., manually, automated thresholding, etc) and how they selected the responsive ROIs (this last point is relevant for the "limit of detection" referred to in line 563). Clarifying the quantification methods is important because it could help to explain the contrast between the results in this manuscript and previous findings that stronger odors elicit bigger responses in KCs in the calyx (Apostolopoulou and Lin, 2020 – Figure S16), where responses were quantified by averaging over all pixels in the calyx.

7. Figure 5C This is a clever way to show that APL odor responses are localized to the part of the calyx innervated by the PNs responding to that odor. The graph is persuasive, but the statistical analysis may not be appropriate. Perhaps the authors do not intend to state that the black curve is significantly different from the blue curve (that would mean only that PA is more strongly activating than FA and MP), but rather that the *slope* of the black curve is *steeper than* the slope of the blue curve? This should be tested using the linear regression analysis tool (for example using a feature in Prism: "Test whether slopes and intercepts are significantly different"). Ideally, the analysis should include a ratio for PA/FA and MP/FA.

8. Figure supp 3E (labelled 3F) – APL response vs. predicted PN response from Hallem and Carlson 2006 data passed through equation from Olsen et al. 2010. Do the odor concentrations match? The authors here use dDL and EtOH at 1:100 and ethyl acetate and pentyl acetate at 1:1000. That's a factor of 10 difference. But Hallem and Carlson only tested odors at factors of 100 difference (10^-2^, 10^-4^, 10^-6^, etc.), so it's not clear how the odors used here correspond to the Hallem and Carlson data. Perhaps an alternative approach is simply to plot the APL response vs. PN response for dDL, MCH and Oct using the data from Figure 2 and Figure supp 3A-D.

9. Figure 3F vs. 4B – why is the KC claw response in the APL ON control in Figure 4B only half the amplitude as in the wildtype response in Figure 3F?

10. Figure supp 2C – For identifying synapses as being on claws vs. being on the dendritic branch, the authors write in the Methods, "The localization of the synapses (e.g., on PN bouton or not) was addressed manually by two separate users in a blind manner" – can they clarify what criteria they used for this manual classification?

11. Line 564-570. The argument against option (ii) (that more MGs respond when APL is blocked because more KCs have back-propagating action potentials) is not convincing. The data indicate only a slight increase in the number of responsive MGs when APL is blocked, which is perfectly consistent with a small number of MGs having a large enough fraction of their 13 KC claws getting extra back-propagating action potentials that their odor response becomes detectable. Perhaps both options should remain on the table.

12. Line 528 "the average strength of postsynaptic responses in KC dendritic claws is homogenous among MGs" – but the histograms show that the response strengths are clearly *heterogeneous* among MGs as there is wide variation between MGs. Consider deleting the words "among MGs and".

13. Line 646 "incomplete silencing might increase KCs' output without affecting sparseness" – this is indeed the case – shown in Lin et al. 2014 Nat Neurosci, Figure 8.

14. Line 555-557 draws an implicit contrast between "the input gain control normalization executed by GABAergic interneurons in the AL (Olsen and Wilson 2008)" vs. "the high-pass filter function performed by inhibitory iPNs at the lateral horn (Parnas et al. 2013)." Actually, these are both the same mechanism – Parnas et al. 2013 notes: "We call this form of inhibition a 'high-pass filter' because it allows high-frequency spike trains to pass (Abbott and Regehr, 2004). Similar phenomena have also been termed input gain control (Olsen et al., 2010) or input division (Mysore and Knudsen, 2012)." Indeed, at the cellular level, both occur by pre-synaptic inhibition.

15. The finding that APL responds more strongly to stronger odors (Figure 2) should be placed in the context of a similar finding in Papadopoulou et al. 2011 (Figure S9).

16. The findings that blocking APL makes the amplitudes of odor responses in KC claws sensitive to stimulus strength (Figure 4) should be placed in the context of similar findings in KC lobes/calyx (Mittal et al. 2020 Nat Comm – Supp Figure 9, Apostolopoulou and Lin 2020 PNAS – Figure S3, S16, Lin et al. 2014 Nat Neurosci – Supp Figure 5), while making note of the different quantification methods.

---

## [Author Response]

Recommendations for the authors.1. The data seem to support the authors conclusions that stronger odor responses in PNs/KCs correlates with stronger APL response and thereby stronger GABAergic feedback. Do correlations seen from comparison of APL response magnitudes for MCH and Oct (Figure 2 and Figure 3) hold true for other odor pair comparisons? If such (APL vs PN/KC) comparisons cannot be made, then the fact that studies of additional odorants are needed for more definitive conclusions should at least be discussed.

Following the suggestion of the Reviewer, we have now correlated directly APL and PN activities in response to Mch, Oct and the weaker odour DL (line 331) The resulting Pearson´s correlation coefficient r = 0.95, supports the view that stronger PN activation levels correspond to stronger APL responses. We agree, nonetheless, with the reviewer that additional odours will be needed to draw a solid correlation and we now state so in the main text (line 330).

2. The main conclusion of the manuscript rests on the findings that in one condition (PN boutons, or KC claws with APL blocked), OCT>MCH, but in another condition (KC claws in wildtype), OCT=MCH. Currently this is shown using p<0.05 vs. p>0.05, but it needs to be shown with either an interaction in a 2-way ANOVA, or directly comparing the difference (Oct-MCH) between the conditions (as they did in comparing the differences Oct-dDL vs. Oct-MCH in Figure 2G vs. 2D) (see Nieuwenhuis et al., Nat Neurosci 2011).

As suggested by the Reviewer, we have compared the Oct-Mch activity difference across conditions, similarly to what we did with the data from Figure 2G-2D. In particular, we compared the difference between Oct- and Mch-induced activities in PN boutons (Figure 3B) vs KC claws (Figure 3F) as well as among PN boutons (Figure 3B) and KC claws of either controls or flies where the output from APL was blocked (Figure 4B). Both comparisons confirmed our conclusion that in PNs Oct > Mch whereas in KC claws (with APL inhibition) Oct ~ Mch. These new results are described in the main text in lines 357-361 and 457-463, respectively, and included in a new Figure 4—figure supplement 1.

3. Early figures (Figure 1A-C) need some labeling to orient non-specialist readers, particularly directional markers (anterior/posterior, medial/lateral), similar to figure 5. Scale bars should also be included on these anatomical figures throughout.

All anatomical figures now include scale bars as well as directional axes.

4. The spatial analysis was done for 5 anterior-posterior zones (Z1-Z5), selected as horizontal slices through the calyx. This seems to make sense for most of the PN terminal fields, but what about the other two axes (medial-lateral, dorsal-ventral)? For instance, some PNs (e.g., the PA-responsive one) traverse more than one A-P slice and have terminal fields in multiple dorsoventral layers.

To be able to address whether APL might display localized responses, we sought to identify odour combinations reliably eliciting spatially separable activation patterns in the calyx. We carefully screened through the anatomical data provided by the hemibrain dataset to identify combinations of PNs that were likely to reliably display segregated boutons. Not all PN subsets display reproducible bouton distribution, and in several cases the bouton distribution is not confined to specific volumes, but rather more distributed in the calyx. In addition, the selected PNs should be activated by odours that display a narrow response. The odour/ PN combinations that we chose fulfil all these criteria *and* fortunately their boutons are segregated along the AP axis, which is our axis of imaging. This particular combination allowed us to distinguish between localized PN bouton responses to different odours in the same animal- and thus to address whether APL responses might be local as opposed to widespread in the entire calyx.

We thus explain our approach more clearly in the text (lines 524-527). We now write:

“Taken together, this combination of odours elicits reproducibly localized responses in the calyx, that can be segregated from one another, allowing to ask whether the response of APL might be local.”

To report whether there might be also some additional asymmetry of APL response to these odours along alternative axes, following the suggestion of the reviewer we re-analysed our APL functional imaging data dividing each optical section (Z1-Z5 in Figure 5B) into three sectors (medial, central, lateral) along the medio-lateral axis. We then summed the GCaMP signal recorded from the same sector in all optical section (e.g., all medial sectors in Z1-Z5), and compared the sum among sectors. We did not observe an asymmetric distribution of responses along the medio-lateral axis.

5. In figure 5, an analysis directly comparing the spatial patterns of APL with PN/KC responses would bolster the evidence for local inhibition in shaping the responses. 5. Does silencing APL alter the regionally-specific responses in KCs? Providing an experimental test would bolster the evidence for a local role of APL modulating these spatial patterns.

We predicted that FA, MP and PA would evoke spatially separable responses in the calyx based on the AL glomeruli they activate and on anatomy data. The Reviewer’s comment prompted us to test the distribution of Kenyon cell functional responses to these specific odours in the calyx. For this, we have now imaged KC claws responses to the same odours used in Figure 5 and quantified the number of responding microglomeruli throughout the entire volume of the calyx. We obtained 5 sections along the P/A axis, as we did for the APL imaging experiment in Figure 5. Though each of the given odours elicited some activity in every imaged plane, we observed a significant difference in the distribution of active microglomeruli (ROI) numbers among the PA-evoked activity versus the other two odours (FA and MP) in the first two sections of our stacks (n=7, multiple t-tests), covering the posterior part of the calyx. Hence, these new data, which we have included in the new supplementary Figure 5A, support the anatomically-based prediction that the three odours elicit responses in separable areas of the calyx. We have now included this experiment in the relative Results section (lines 519-524) and added the data in the new Figure 5—figure supplement 1A.

Regarding the second point, we reckon that the reviewer hints at the spatial distribution of KC responses reported in Figure 5—figure supplement 1A. Importantly, we do not suggest that these regional patterns are determined by APL. In fact, they correlate with the anatomical distribution of PN boutons, which we think is the main driver of KC response regional distribution. Conversely, we propose that APL activation is regionalized – due to the distribution of PN inputs. As we mention in our discussion, local inhibition, as opposed to global inhibition, is suggested to expand the dynamic range of inputs able to activate KCs (Ray, Aldworth, and Stopfer 2020).

6. The authors quantify PN bouton and KC claw responses by selecting only ROIs that responded to the odor. (line 303-4: "calculated per calyx the average peak response among the boutons showing calcium transients") However, sometimes they refer to "average bouton response" or "average microglomerular postsynaptic response" which could reasonably be interpreted to mean the average including non-responsive boutons/claws. This point should be clarified either by changing the phrasing or adding more explanations of the quantification. It's also not clear if the responses in Figure 3B, 3F, 4B are quantified for the same ROIs across both odors, or if for each odor, only the ROIs responsive to that particular odor were used. Could the authors be more explicit about their quantification methods? In the Methods section, they should explain how they identified bouton/claw ROIs (e.g., manually, automated thresholding, etc) and how they selected the responsive ROIs (this last point is relevant for the "limit of detection" referred to in line 563). Clarifying the quantification methods is important because it could help to explain the contrast between the results in this manuscript and previous findings that stronger odors elicit bigger responses in KCs in the calyx (Apostolopoulou and Lin, 2020 – Figure S16), where responses were quantified by averaging over all pixels in the calyx.

Throughout this work, in every experiment in which active boutons or postsynaptic responses in microglomeruli were analysed, responsive ROIs were automatically identified via the custom-made “Cell Tracking by calcium” imageJ macro, written by our DZNE Image and Data Analysis Facility (IDAF).

Briefly, given the expected ROI diameter (for our experiments, this value was set to 5 μm, in agreement with Leiss et al. 2009 and Kremer et al., 2010), this macro normalizes the whole dataset to the minimum image, obtained via ImageJ´s “Z-project” function, and thresholds the resulting data via a method of choice (default and used in this work: Otsu method, often used for analysis of dynamic fluorescence imaging data as in Kaifosh et al., 2014) in order to identify the active regions within a particular odour exposure. Hence, a ROI will be labelled as active only if its pixel intensity variation is above threshold and if its size matches with the expected ROI diameter value. Additional filters are automatically applied before this segmentation step, in order to reduce detectors noise, scattered light or artifacts due to a sub-optimal image registration. A detailed PDF explaining the principle of this macro, together with the source file via which the macro can be run on ImageJ, are provided in the dryad folder related to our manuscript (https://doi.org/10.5061/dryad.bk3j9kdd1, though to our understanding the link seems inaccessible until publication). We have included an explanation of the segmentation process as well as the calculation of the average peak response value in our Material and Methods section (lines 817-828).

Nevertheless, the output of the macro is a mask, which is then re-applied to the original raw data in order to extract the intensity value over time of each ROI and calculate ΔF/F_0_% and ΔF/F_0_% MAX values per each responding microglomerulus (see Author response image 1).

**Author response image 1. sa2fig1:** Cell Tracking by calcium macro. Example of detection of active ROIs during odour exposure. Left, original raw file. Centre, mask of active ROIs obtained via the “Cell Tracking by calcium” ImageJ macro. Right, the obtained mask is then used to extract the pixel intensity values over time of the regions labelled as active ROI. Afterwards, ΔF/F_0_% and ΔF/F_0_% MAX values per each ROI are calculated and analysed.

To answer the Reviewer´s specific point, the average bouton response and the average microglomerular postsynaptic response in each odour exposure trial were calculated only among ROIs that have been classified as active. Hence, as different odours lead to the activation of different PNs/KCs, the responses quantified in Figure 3B, 3F and 4B derive from different ROIs. We have made this clearer by rephrasing the various “average peak response” in the manuscript (lines 317; 321; 350; 372 and 434).

Finally, concerning our results and previous findings on KC responses in the calyx (Apostolopoulou and Lin, 2020 – Figure S16), we agree with the Reviewer that the observed contrast is most likely generated by the two different means of quantification: by averaging over all pixels in the calyx, fluorescence out of an unknown number of active microglomeruli is collected and grouped. Hence, differences in the strength of odour-evoked activity in the aforementioned work could be explained by an elevated/reduced number of active microglomeruli included in the calycal area measured for the analysis. We now point to this in the main text (lines 348-350).

7. Figure 5C This is a clever way to show that APL odor responses are localized to the part of the calyx innervated by the PNs responding to that odor. The graph is persuasive, but the statistical analysis may not be appropriate. Perhaps the authors do not intend to state that the black curve is significantly different from the blue curve (that would mean only that PA is more strongly activating than FA and MP), but rather that the slope of the black curve is steeper than the slope of the blue curve? This should be tested using the linear regression analysis tool (for example using a feature in Prism: "Test whether slopes and intercepts are significantly different"). Ideally, the analysis should include a ratio for PA/FA and MP/FA.

We have now performed the slope comparison suggested by the Reviewer, which confirmed that the slope of the two curves is significantly different (p=0.0004). We have added this information to the legend of Figure 5C (lines 566-568) and to the legend of Figure 5—figure supplement 1B (Lines 583-585). Moreover, we have analysed the ratio for PA/FA vs MP/FA, which showed the same trend (p=0.0003), and plotted the results in a new Figure 5—figure supplement 1B. In the main text, we wrote (lines 540-542):

“Of notice, we observed the same trend when comparing the ratio of PA to FA against the ratio of MP to FA (Figure 5—figure supplement 1B).”

8. Figure supp 3E (labelled 3F) – APL response vs. predicted PN response from Hallem and Carlson 2006 data passed through equation from Olsen et al. 2010. Do the odor concentrations match? The authors here use dDL and EtOH at 1:100 and ethyl acetate and pentyl acetate at 1:1000. That's a factor of 10 difference. But Hallem and Carlson only tested odors at factors of 100 difference (10^-2^, 10^-4^, 10^-6^, etc.), so it's not clear how the odors used here correspond to the Hallem and Carlson data. Perhaps an alternative approach is simply to plot the APL response vs. PN response for dDL, MCH and Oct using the data from Figure 2 and Figure supp 3A-D.

The recorded APL activity in response to odour stimulation positively correlates with the predicted PN responses to stimulation with the same odours at 10^-4^ and 10^-2^ dilutions derived from the Hallem and Carlson data. Nonetheless, the actual concentration used in our experiments lays in-between these values and is missing from the database: we are sorry for this mistake. As suggested by the Reviewer, we have now correlated APL activity with the PN experimental data shown in Figure 3—figure supplement 2A-D (line 331). As mentioned above, the resulting Pearson´s correlation coefficient r = 0.95 supports the view that stronger PN activation levels correspond to stronger APL responses.

9. Figure 3F vs. 4B – why is the KC claw response in the APL ON control in Figure 4B only half the amplitude as in the wildtype response in Figure 3F?

A temperature-dependent decrease in GCaMP responses of control groups was observed in a very similar experimental setup in Lin et al. 2014 (e.g. Figure 4d), and described as a possible temperature dependence of calcium binding and/or subsequent conformational changes in the GCaMP3, cellular calcium dynamics and/or Kenyon cell spike rates. We have now mentioned this phenomenon and added the Lin 2014 reference in our main text (lines 440-444)

10. Figure supp 2C – For identifying synapses as being on claws vs. being on the dendritic branch, the authors write in the Methods, "The localization of the synapses (e.g., on PN bouton or not) was addressed manually by two separate users in a blind manner" – can they clarify what criteria they used for this manual classification?

The position of APL synapses onto KC was addressed via visual judgment of the 3-dimensional structure of the KC quantified. As the Hemibrain dataset does not distinguish between claws and dendritic branches yet, a computational approach is not possible for this analysis. Nevertheless, as “meshes” represent filled neurons and not simply skeletons, this distinction was recognisable visually, as confirmed by the low variability in the quantification between the two blind users. Additionally, in case of particularly tricky cases, both PN inputs and receiving KCs 3D reconstructions were plotted, as shown in Author response image 2, allowing for an easy identification of the microglomerular unit from the rest. To help readers understand this process, we have extended our material and methods section regarding this quantification (lines 737-741). Now we write:

“In particular, the localization of an APL-to-PN or APL-to-KC synapse was judged by looking at the filled 3D render of the neuron of interest. In case of uncertainties, both PN inputs and receiving KCs were plotted in the same image in order to directly identify the microglomerular structure.”

**Author response image 2. sa2fig2:** Distinction between synapses on KC dendritic endings (claws) vs not on claws. In this example, one PN bouton providing input to a KC of interest is visualized. The KC dendritic specialization involved in the microglomerular structure (i. e. the claw) is defined by the fact that it is in direct contact with the bouton. The APL presynaptic site marked by an arrowhead is located on a claw (note the contact between the green claw and the red bouton). In contrast, the localization of the APL presynaptic site marked by an arrow is not on a claw but along the dendritic branch.

11. Line 564-570. The argument against option (ii) (that more MGs respond when APL is blocked because more KCs have back-propagating action potentials) is not convincing. The data indicate only a slight increase in the number of responsive MGs when APL is blocked, which is perfectly consistent with a small number of MGs having a large enough fraction of their 13 KC claws getting extra back-propagating action potentials that their odor response becomes detectable. Perhaps both options should remain on the table.

We reconsidered our view, especially in light of recent evidence of co-connectivity of certain PN/KC subsets, and agree that both explanations could be possible. However, given the length of the discussion, we decided to cut this part in the revised manuscript.

12. Line 528 "the average strength of postsynaptic responses in KC dendritic claws is homogenous among MGs" – but the histograms show that the response strengths are clearly heterogeneous among MGs as there is wide variation between MGs. Consider deleting the words "among MGs and".

We are sorry that our wording was unclear. It is correct that the ΔF/F_0_ peaks of individual MGs responding to a particular odour trial are heterogeneous. What is homogeneous is the distribution of ΔF/F_0_ peaks in postsynaptic MG responses among different odour representations in the presence of APL inhibition, which is indicated by the overlapping histograms in Figure 3G, 4B and Figure 3—figure supplement 2G. This overlap was disrupted in the absence of APL (e.g., Figure 4F). To make this clearer, we have modified the sentence in the following way (lines 597-599):

“[…] the average strength and the distribution of postsynaptic responses in KC dendritic claws become more similar across different odour representations”.

13. Line 646 "incomplete silencing might increase KCs' output without affecting sparseness" – this is indeed the case – shown in Lin et al. 2014 Nat Neurosci, Figure 8.

We had indeed cited that publication just one sentence below, but for clarity, we have now modified this sentence slightly and added the reference (lines 709-712):

“[…] incomplete silencing via RNAi increases KC output without affecting sparseness, whereas blocking APL output via shibire^ts^ leads to large, overlapping odour representations and impaired olfactory discrimination (A. C. Lin et al. 2014).”

14. Line 555-557 draws an implicit contrast between "the input gain control normalization executed by GABAergic interneurons in the AL (Olsen and Wilson 2008)" vs. "the high-pass filter function performed by inhibitory iPNs at the lateral horn (Parnas et al. 2013)." Actually, these are both the same mechanism – Parnas et al. 2013 notes: "We call this form of inhibition a 'high-pass filter' because it allows high-frequency spike trains to pass (Abbott and Regehr, 2004). Similar phenomena have also been termed input gain control (Olsen et al., 2010) or input division (Mysore and Knudsen, 2012)." Indeed, at the cellular level, both occur by pre-synaptic inhibition.

With this phrase, we wanted to stress that input gain controls are present at different levels of the fly olfactory pathway, but we can see how our writing might have been misleading. Following the reviewer’s suggestion for more clarity, we have rephrased this sentence and lines 626-629 now read:

“Indeed, input gain control normalization has been described for GABAergic interneurons in the AL (Olsen and Wilson 2008) as well as for inhibitory iPNs at the lateral horn (Parnas et al. 2013).”

15. The finding that APL responds more strongly to stronger odors (Figure 2) should be placed in the context of a similar finding in Papadopoulou et al. 2011 (Figure S9).

The experiments carried out by Papadopoulou deal with raising concentrations of the same odour, that led to increasing depolarization of APL. In Figure 2, we compare different odours that activate different subsets of PNs, eliciting stronger or weaker responses. Following the Reviewer’s advice to discuss our findings in the context of the Papadopoulou data, we have now added the following sentence in the discussion, lines 629-633:

“APL and its homolog GGN in the locust showed increased depolarization in response to increasing odour concentration (Papadopoulou et al. 2011). This, combined with our findings, suggests that the normalization performed by APL might be acting not only across stimuli identities, but also among concentrations of the same stimulus.”

16. The findings that blocking APL makes the amplitudes of odor responses in KC claws sensitive to stimulus strength (Figure 4) should be placed in the context of similar findings in KC lobes/calyx (Mittal et al. 2020 Nat Comm – Supp Figure 9, Apostolopoulou and Lin 2020 PNAS – Figure S3, S16, Lin et al. 2014 Nat Neurosci – Supp Figure 5), while making note of the different quantification methods.

We have now added a sentence pointing to the supporting results described in Mittal 2020, Apostolopoulou 2020 and Lin 2014. The text now reads (lines 463-466):

“Remarkably, KC responses recorded in the MB lobes were also higher and more diverse for different odours in the absence of APL inhibition (Mittal et al. 2020; Apostolopoulou and Lin 2020; A. C. Lin et al. 2014).”

Regarding the difference in methods utilized in Apostolopoulou and Lin 2020 or in the present work to quantify the calycal response, please see also our answer to point 6 and the newly added sentence included in the main text at lines 348-350.